# Neoadjuvant Carboplatin/Paclitaxel versus 5-Fluorouracil/Cisplatin in Combination with Radiotherapy for Locally Advanced Esophageal Squamous Cell Carcinoma: A Multicenter Comparative Study

**DOI:** 10.3390/cancers14112610

**Published:** 2022-05-25

**Authors:** Xing Gao, Ping-Chung Tsai, Kai-Hao Chuang, Chu-Pin Pai, Po-Kuei Hsu, Shau-Hsuan Li, Hung-I Lu, Joseph Jan-Baptist van Lanschot, Yin-Kai Chao

**Affiliations:** 1Division of Thoracic Surgery, Chang Gung Memorial Hospital-Linkou, Chang Gung University, Taoyuan 333, Taiwan; gaoxing1993@yahoo.com; 2Department of Surgery, Erasmus Medical Center, 3015GD Rotterdam, The Netherlands; j.vanlanschot@erasmusmc.nl; 3Division of Thoracic Surgery, Department of Surgery, Taipei Veterans General Hospital, Taipei 112, Taiwan; pctsai9@vghtpe.gov.tw (P.-C.T.); bcbjohnny@gmail.com (C.-P.P.); hsupokuei@yahoo.com.tw (P.-K.H.); 4Division of Thoracic Surgery, Chang Gung Memorial Hospital-Kaohsiung, Chang Gung University, Kaohsiung 833, Taiwan; maetin221@gmail.com (K.-H.C.); lee0624@cgmh.org.tw (S.-H.L.); luhungi@cgmh.org.tw (H.-I.L.)

**Keywords:** esophageal cancer, esophageal squamous cell carcinoma, neoadjuvant, chemoradiotherapy

## Abstract

**Simple Summary:**

The most beneficial neoadjuvant chemoradiotherapy for Asian patients with esophageal squamous cell carcinoma remains uncertain. Using propensity score matching by inverse probability of treatment weighting to balance the baseline variables, the neoadjuvant carboplatin/paclitaxel (CROSS) regimen versus the cisplatin/5-fluorouracil (PF) regimen in combination with 41.4–50.4 Gy of radiotherapy were compared. We found that Taiwanese patients treated with the CROSS regimen (Carboplatin + Paclitaxel + 41.4–45.0 Gy) had less treatment-related complications and more favorable survival figures. Collectively, these results suggest that CROSS is safe and effective.

**Abstract:**

Background: The most beneficial neoadjuvant chemoradiotherapy (nCRT) combination for esophageal squamous cell carcinoma (ESCC) in Asia remains uncertain. Herein, we compared the neoadjuvant carboplatin/paclitaxel (CROSS) regimen versus the cisplatin/5-fluorouracil (PF) regimen in combination with 41.4–50.4 Gy of radiotherapy. Methods: Patients were stratified according to their nCRT regimen: CROSS + 41.4–45.0 Gy (CROSS), PF + 45.0 Gy (PF4500) or PF + 50.4 Gy (PF5040). Propensity score matching by inverse probability of treatment weighting (IPTW) was used to balance the baseline variables. Results: Before IPTW, a total of 334 patients were included. The lowest chemotherapy completion rate was observed in the PF5040 group (76.2% versus 89.4% and 92.0% in the remaining two groups, respectively). Compared with CROSS, both PF groups showed more severe weight loss during nCRT and a higher frequency of post-esophagectomy anastomotic leaks. The use of PF5040 was associated with the highest rate of pathological complete response (45.3%). While CROSS conferred a significant overall survival benefit over PF4500 (hazard ratio [HR] = 1.30, 95% CI = 1.05 to 1.62, *p* = 0.018), similar survival figures were observed when compared with PF5040 (HR = 1.17, 95% CI = 0.94 to 1.45, *p* = 0.166). Conclusions: The CROSS regimen conferred a significant survival benefit over PF4500, although the similar survival figures were similar to those observed with PF5040. Considering the lower incidences of severe weight loss and post-esophagectomy anastomotic leaks, CROSS represents a safe and effective neoadjuvant treatment for Taiwanese patients with ESCC.

## 1. Introduction

Treatment options for esophageal cancer (EC) have rapidly expanded from surgery alone to multimodal approaches comprising surgery, chemotherapy (CT) and/or radiotherapy (RT) [1,2]. Currently, neoadjuvant chemoradiotherapy (nCRT) followed by surgery is the mainstay of treatment for patients with locally advanced EC [3,4,5]. High-quality evidence from previous clinical trials indicated that both carboplatin plus paclitaxel and cisplatin plus 5-fluorouracil (PF) based regimens are significantly effective at improving survival of patients with EC [6,7,8]. As for RT, the recommended neoadjuvant dose is between 41.4–50.4 Gy [6,9].

Carboplatin plus paclitaxel with concurrent 41.4 Gy RT (CROSS) is the current standard of care for neoadjuvant chemoradiotherapy (nCRT) in many practices. The relatively low toxicity profile of the CROSS regimen compared to PF while maintaining the same efficacy has made CROSS widely popular as nCRT in Western regions, mainly in response to esophageal adenocarcinoma [10,11,12]. While the subset of patients with ESCC in the CROSS trial showed promising findings (i.e., pathologically complete response [pCR] rate = 49%; median survival = 81.6 months), replication of these findings in Asia has not been successful [13,14,15].

In Asia, more than 90% of all EC diagnosed are esophageal squamous cell carcinoma (ESCC). Not only do risk factors differ from Western regions (i.e., hot drinks, foods containing N-nitroso compounds), varying responses to anticancer therapies between Asian and Caucasian patients has also been linked to varying frequencies of genetic polymorphisms [16,17]. Furthermore, neoadjuvant treatment is reserved to patients with more advanced malignancies (cT3-4aN1-3M0) in Asia [18]. The efficacy of CROSS in advanced stages of ESCC has recently been called into question by a study from Hong Kong—which showed an unfavorable survival trend for patients who received CROSS versus those who had been treated with PF [13]. In addition, traditional PF schemes with 41.4–50.4 Gy of RT are still extensively used for nCRT in Asian patients with EC [3].

Therefore, we aimed to compare CROSS to the two most common nCRT combinations given in Taiwan—i.e., PF plus 45.0 Gy of RT (PF4500) and PF plus 50.4 Gy of RT (PF5040)—in terms of overall survival (OS), pathological complete response (pCR), and treatment-related complications. Compared to CROSS, patients who received PF regimens were expected to achieve comparable OS despite a higher burden of treatment-related complications. Propensity score matching with stratification by inverse probability of treatment weighting (IPTW) for nine baseline variables were used to balance the baseline variables between the treatment groups.

## 2. Materials and Methods

The study protocol was approved by the Chang Gung Memorial Hospital institutional review board (approval number: 202100729B0). The requirement for written patient informed consent was waived based on the study design.

### 2.1. Study Participants

Participants were recruited from three high-volume (defined as >20 esophagectomies per year) medical centers in Taiwan [19] (Chang Gung Memorial Hospital-Linkou, Taipei Veterans General Hospital, Chang Gung Memorial Hospital-Kaohsiung). All patients diagnosed with ESCC who received nCRT as upfront treatment between 2010 and 2018 were included. Exclusion criteria comprised (1) other malignancies identified in the five years preceding the diagnosis of ESCC or multiple tumors at diagnosis, (2) distant metastases according to the American Joint Committee on Cancer (AJCC) Staging Manual, eighth edition, (3) cervical ESCC, and (4) an Eastern Cooperative Oncology Group performance status ≥2. Eligible patients were divided into three groups according to the nCRT regimen they have undergone (CROSS versus PF5040 versus PF4500).

### 2.2. Neoadjuvant Chemoradiotherapy and Surgery

Patients in the CROSS group received six weekly cycles of intravenous carboplatin (area under curve = 2 mg/mL/min) and paclitaxel (50 mg/m^2^ body surface area) administered on the first day of each week. Patients in the PF groups received two cycles of cisplatin (75 mg/m^2^) on day 1 and day 29 combined with continuous infusion of 5-fluorouracil (1000 mg/m^2^) per day for 4 days starting from day 1 and day 29, respectively. Neoadjuvant RT—which consisted of external-beam photon radiation given in fractions (dose per fraction: 1.8 Gy) five days per week—was generally started concurrently on the first day of CT. Patients were scheduled to receive a total of 23 (41.4 Gy), 25 (45.0 Gy), or 28 (50.4 Gy) fractions. In all centers, the gross tumor volume comprised the primary tumor area and all the adjacent suspected lymph nodes. When distant nodal involvement outside of the maximum tolerated radiation field was suspected, patients received an interrupted radiation field. Different RT techniques (i.e., volumetric modulated arc therapy, three-dimensional conformal RT, or intensity-modulated RT) were used throughout the study period. The standard surgical approach consisted of a transthoracic esophagectomy with intrathoracic gastric tube reconstruction (Ivor Lewis procedure) or cervical anastomosis (McKeown procedure). All the study patients underwent two-field lymph node dissection. Cervical lymphadenectomy was performed in selected patients who showed evidence of residual disease in the cervical area.

### 2.3. Definitions

The presence of comorbidities was assessed using the age-adjusted Charlson’s comorbidity index [20]. CT was considered completed after at least five (CROSS) or two (PF4500 or PF5040) cycles. In accordance with the published literature, weight loss during nCRT of more than 10% of the initial body weight was considered as severe [21,22]. The occurrence of postoperative pneumonia was investigated using the Revised Uniform Pneumonia Score [23]. Anastomotic leaks and the presence of chylothorax were assessed using the Esophagectomy Complication Consensus Group scoring system [24]. The Clavien–Dindo criteria were applied to determine the severity of postoperative complications [25]. The criterion for pCR was the absence of malignant cells in all of the resected pathology specimens (both primary tumor and lymph nodes). Tumor regression grade (TRG) was assigned according to a modified Mandard score, as follows: TRG 1, no viable residual tumor cells (VRTCs); TRG 2, single or small groups of VRTCs; TRG 3, VRTCs outgrown by fibrosis, and TRG4, fibrosis outgrown by VRTCs or extensive residual cancer [26]. Overall survival (OS) was defined as the time elapsed from the first day of nCRT to the date of death. Progression-free survival (PFS) was defined as the time elapsed from the initiation of nCRT to the day of disease progression, including the day of surgery (when resections were incomplete) or the day of progression during follow-up (for patients who did not undergo surgery).

### 2.4. Statistical Analysis

To account for potential differences in baseline variables between multiple treatment groups (CROSS versus PF5040 versus PF4500), we applied the inverse probability of treatment weighting (IPTW) method based on the generalized multiple propensity scores (PSs) [27,28]. PSs were estimated using the generalized boosted model (GBM) based on 50,000 regression trees; this machine learning method has been shown to outperform simple logistic regression models in most scenarios [29]. We assessed the existence of an intergroup balance before and after the application of the GBM-IPTW method by calculating the maximum absolute standardized difference (MASD) between pairs; in these analyses, a value <0.2 indicates a negligible difference [29]. Among the baseline variables listed in Table 1, those identified as independent predictors of survival were subjected to PS matching [30]. In the GBM-IPTW adjusted cohort, OS and PFS curves were plotted using the Kaplan–Meier method. To evaluate the associations between risk factors and clinical outcomes, we performed a Cox proportional hazard regression. Results are expressed as hazard ratios (HRs) and 95% confidence interval (CIs). In pre-specified subgroup analyses, we examined the differences in terms of OS between the CROSS and PF5040 groups. One-way analysis of variance and Fisher’s exact tests were used to analyze continuous and categorical data, respectively. A *post-hoc* Bonferroni’s correction was applied when the overall test was statistically significant. All calculations were performed using SAS, version 9.4 (SAS Institute, Cary, NC, USA) with the TWANG macro for estimating GBM-IPTW. Two-tailed *p* values < 0.05 were considered statistically significant.

## 3. Results

### 3.1. Baseline Characteristics

Three-hundred thirty-four patients were eligible for inclusion (CROSS *n* = 124 versus PF5040 *n* = 105 versus PF4500 *n* = 105). PS analyses stratified according to nine baseline variables were employed to account for potential confounders (Table 1). Baseline characteristics on diagnosis were comparable before IPTW, the only exceptions being clinical N stage and the years of nCRT. In the GBM-IPTW-adjusted cohort, a better balance was achieved (all MASD values < 0.20), except for the years of nCRT (MASD = 0.29). This was expected since the CROSS regimen has been introduced more recently.

### 3.2. Neoadjuvant Chemoradiotherapy Related Outcomes

Neoadjuvant chemoradiotherapy-related outcomes in the preoperative period before and after the application of the GBM-IPTW method are shown in Table 2 and Appendix A. While the CT completion rate was significantly lower in the PF5040 group (76.2%), no significant differences were observed between the CROSS and PF4500 groups (89.4% and 92.0%, respectively). A significantly higher percentage of patients in the PF4500 (15.8%) and PF5040 (16.5%) groups showed severe weight loss compared with the CROSS group (8.5%). The resection rates after nCRT varied from 69.2% to 76.6% but did not show significant intergroup differences. The main reasons that led to avoidance of surgery were disease progression (8.8 to 15.6%) and patient’s refusal (7.3 to 12.7%).

### 3.3. Perioperative and Pathology Outcomes

The perioperative outcomes and pathological findings observed in the GBM-IPTW adjusted cohort are summarized in Table 3, whereas data for the original cohort are reported in Appendix A. While the use of thoracoscopic surgery was similar in the three groups (>95% of all cases), a higher number of patients in the CROSS group received a laparoscopic approach for reconstruction. On analyzing postoperative morbidity, patients in the PF4500 (26.6%) group had a higher incidence of pulmonary complications compared with the CROSS (14.1%) and the PF5040 (11.3%) groups. Similarly, there was a higher incidence of chylothorax in the PF5040 group compared with the CROSS group (6.7% versus 1.4%, respectively; *p* < 0.05). Compared to CROSS, both the PF4500 and PF5040 regimens were associated with significantly higher rates of anastomotic leaks. When applying the Clavien–Dindo criteria, we found no significant differences in terms of major complications and postoperative death rates among the three regimens. While the three groups did not differ significantly in terms of R0 resection rates, a stepwise decrease in the number of dissected nodes was observed in the CROSS, PF4500, and PF5040 groups (27.3 versus 23.0 versus 18.4, respectively). Patients in the PF5040 group exhibited a significantly higher pCR rate (45.3%) compared with those in the PF4500 and CROSS groups (31.6% and 29.5%, respectively, *p* < 0.05).

### 3.4. Survival Analysis

In an intention-to-treat analysis, patients in the CROSS group showed a significantly better OS compared with the PF4500 group (HR = 1.30, 95% CI = 1.05 to 1.62, *p* = 0.018) but not with the PF5040 group (HR = 1.17, 95% CI = 0.94 to 1.45, *p* = 0.166) (Figure 1a). In the CROSS, PF5040, and PF4500 groups, the median survival times (MST) were 28.5 months (95% CI = 18.6 to 62.1 months), 20.9 months (95% CI = 15.8 to 28.8 months), and 19.5 months (95% CI = 14.6 to 34.1 months), respectively. The corresponding 5-year OS rates were 40.4% (95% CI = 30.8 to 50.1%), 32.4% (95% CI = 21.6 to 43.2%), and 30.1% (95% CI = 19.4 to 40.9%), respectively. Similar figures were observed for PFS rates (Figure 1b).

### 3.5. Survival Analysis in the CROSS and PF5040 Groups

The results of subgroup analyses are summarized in Table 4, which depicts the survival of patients who received the CROSS or PF5040 regimen. No statistically significant favorable effects were observed in the CROSS group after stratification for several prespecified variables—the only exception being completion of CT (*p* for interaction < 0.05). Among patients who could not complete CT, significantly lower OS figures were observed when CT of the PF5040 regimen could not be completed (HR = 3.81; 95% CI = 2.11 to 6.89); however, this was not the case for patients who did not complete the CT course of the CROSS regimen (HR = 0.88; 95% CI = 0.69 to 1.13).

## 4. Discussion

Prior investigations focusing on the safety and efficacy of the CROSS regimen have been mainly conducted in Western countries and the results might not be generalizable to different ethnicities. Additionally, Asian patients are generally offered nCRT only in the presence of advanced tumors (cT3-4aN1-3M0). This practice is different from that implemented in Western countries, where all locally advanced resectable tumors (cT1N1-3M0/cT2-4aN0-3M0) are treated with nCRT prior to surgery [16,17,18]. To our knowledge, no prospective study in Asia has addressed the effectiveness of CROSS, and retrospective studies in the field have been inconclusive [13].

The current multicenter retrospective study compared the CROSS regimen with two commonly applied schemes (PF + 45.0 Gy and PF + 50.4 Gy) in Taiwan [3,6]. Our findings indicate that the use of CROSS was associated with significantly better OS and PFS over PF4500, with similar survival figures when compared to PF5040. Additionally, the incidence rates of severe weight loss and postoperative anastomotic leaks were lower for patients who were treated with CROSS. Overall, the CROSS regimen should be considered as a safe and effective neoadjuvant treatment for Asian patients with ESCC.

Previous findings have already supported the better tolerability of CROSS compared to PF schemes [10]. PF regimens have been associated with a higher frequency of grade III adverse events in definitive CRT settings and more pronounced weight loss in neoadjuvant settings [11,31]. In addition, severe weight loss has been consistently linked to a higher incidence of anastomotic leaks in patients who had undergone gastrointestinal resections and is consistent with our findings in the PF4500 and PF5040 groups [21,32,33].

However, the relatively low nCRT efficacy of patients in our CROSS group was rather unexpected. The pCR rate was 49% in the ESCC subset of the original CROSS trial but only 29.5% in our cohort [6]. This difference may be attributed to a higher number of patients with advanced disease in our cohort (cN+ rate: 96.6% versus 65% in the original CROSS trial) as well as to different EC biology, suggesting the ethnicity may have an impact on pCR rates. Interestingly, a similar pCR rate after CROSS (24.5%) was reported in a recent study conducted in Hong Kong [13]. Although pCR rates were consistent with our current results, this study showed an unfavorable trend for CROSS compared to PF (MST 32.7 versus 16.7 months, respectively, *p* = 0.083). Appendix A compares the key findings of the two investigations. In the Hong Kong study, however, the two cohorts were treated in different time periods (PF 2002−2012 versus CROSS 2012−2019). This may represent a significant confounder both in terms of different patient characteristics over time but also in terms of treatment aggressiveness. As the authors state [13], patients with borderline unresectable tumors might erroneously have been classified as neoadjuvant candidates to receive “conversion therapy”. This can in part explain the unfavorable survival outcomes, the high disease progression rate (17%), and the low resection rate (69%) observed in their CROSS group.

Patients in our PF5040 group had the highest pCR rate (45.3%), but similar OS and PFS when compared with CROSS. This unexpected finding might be attributed to a suboptimal therapeutic delivery of PF5040 due its higher toxicity profile. Suboptimal application of nCRT has been previously associated with less favorable OS and PFS in other solid malignancies (e.g., rectal and bladder tumors) [34,35]. While patients in the PF5040 group received high RT doses (Table 2), the rate of CT completion was significantly lower than those observed in patients who received CROSS or PF4500. Furthermore, the detrimental effect of suboptimal therapy appeared more profound in the PF5040 group. In our subgroup analyses, patients who did not complete PF5040 had significantly poorer OS rates than those who were unable to complete the CROSS regimen (Table 4). Besides a suboptimal completion of CT, a higher burden of postoperative complications has an adverse prognostic significance. This may offer an explanation for our findings in the PF4500 group (Table 3) [36,37]. Taken together, these results suggest that clinicians should thoroughly assess the patient’s ability to tolerate a regimen before allocation. This should ideally take place at baseline in a multidisciplinary context to avoid irreversible decisions that—under certain circumstances—could reduce a patient’s survival chances.

While our study enrolled the largest cohort of Asian patients with ESCC treated with CROSS or PF during the same time period, several limitations need to be considered. First, the choice to give CROSS or PF depended on availability and physician preference. However, the costs of the CROSS regimen are not entirely covered by the Taiwanese national insurance system; this could have led to a selection bias where patients who ultimately received CROSS had better financial capacities. Second, our original CROSS cohort had more advanced clinical nodal status compared to the PF groups (Table 1). However, the results following application of IPTW revealed that the outcomes of patients with cN0 and cN1 in the CROSS cohort were amplified. This may have resulted in better survival figures by neutralizing a possible disadvantage of CROSS in patients with a higher burden of nodal disease. Third, patients in the CROSS group had a significantly higher nodal yield, which is known to affect survival outcomes [38,39]. It would have been interesting to investigate the potential reasons underlying this phenomenon (e.g., different surgical approaches and/or heterogeneous RT doses); however, as the study had a retrospective nature, we were unable to analyze this variable. Finally, we did not conduct a formal sample size calculation; therefore, randomized controlled trials are necessary to confirm our findings and to evaluate the impact of different nCRT regimens in terms of survival and complications for Asian patients with ESCC.

## 5. Conclusions

Our findings indicate that, in Taiwanese patients with ESCC, the use of the CROSS regimen for nCRT is associated with a significantly better survival compared with PF4500 and similar survival figures compared with PF5040. Considering the lower incidences of severe weight loss and post-esophagectomy anastomotic leaks, CROSS represents a safe and effective neoadjuvant treatment in Asian patients with ESCC.

## Figures and Tables

**Figure 1 cancers-14-02610-f001:**
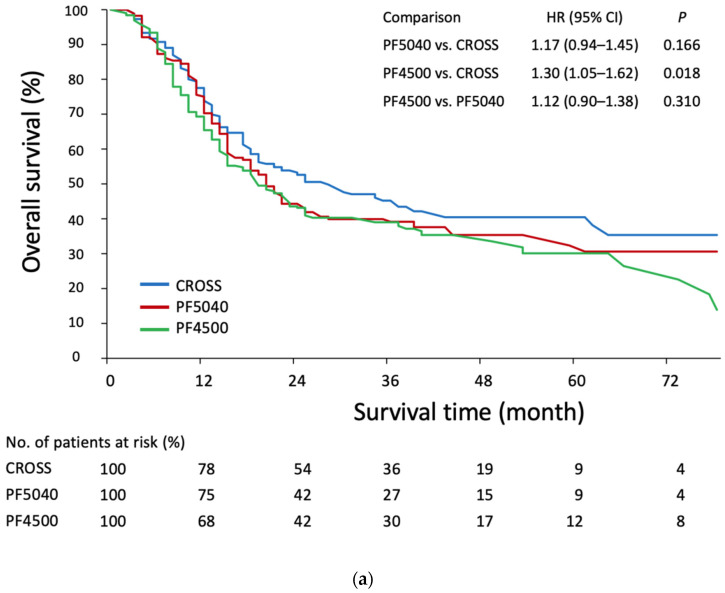
(**a**). Kaplan–Meier estimates of overall survival in the GBM-IPTW adjusted cohort, stratified according to the three nCRT protocols. (**b**) Kaplan–Meier estimates of progression-free survival in the GBM-IPTW adjusted cohort, stratified according to the three nCRT protocols. Abbreviations: GBM-IPTW, generalized boosted modeling-inverse probability of treatment weighting; HR, hazard ratio; CI, confidence interval.

**Table 1 cancers-14-02610-t001:** Baseline characteristics of the three study groups before and after GBM-IPTW.

Variable	Before IPTW (Original Cohort)					After IPTW				
	CROSS (*n* = 124)	PF5040 (*n* = 105)	PF4500 (*n* = 105)	*p* Value	MASD	CROSS	PF5040	PF4500	*p* Value	MASD
Male sex	118 (95)	92 (88)	100 (95)	0.061	0.30	94.6	90.5	93.5	0.17	0.16
Age, years	57.6 ± 8.9	57.3 ± 9.1	55.4 ± 9.9	0.19	0.23	57.3 ± 9.1	56.7 ± 9.1	56.4 ± 9.1	0.53	0.10
* BMI, kg/m^2^	22.4 ± 3.6	22.1 ± 3.2	22.4 ± 3.7	0.81	0.09	22.4 ± 3.6	22.0 ± 3.1	22.6 ± 3.8	0.095	0.17
ACCI score				0.050	0.23				0.7	0.07
0	14 (11)	10 (10)	18 (17)			13.5	11.4	13.5		
1−2	78 (63)	53 (51)	62 (59)			61.1	58.0	58.5		
≥3	32 (26)	42 (39)	25 (24)			25.4	30.6	28.1		
Clinical T stage				0.12	0.17				0.38	0.15
1	1 (0.8)	3 (3)	1 (1)			1.1	2.9	1.6		
2	17 (14)	11 (11)	24 (23)			15.7	11.2	16.0		
3	101 (82)	88 (84)	79 (75)			80.3	83.9	80.9		
4	5 (4)	3 (3)	1 (1)			2.9	2.0	1.5		
Clinical N stage				<0.001	0.30				0.26	0.14
0	3 (2)	7 (7)	10 (10)			3.4	6.0	6.6		
1	36 (29)	61 (58) ^a^	46 (44)			38.6	46.6	42.4		
2	68 (55)	29 (28) ^a^	37 (35) ^a^			46.9	38.0	41.3		
3	17 (14)	8 (8)	12 (11)			11.1	9.4	9.6		
Clinical stage				0.12	0.17				0.78	0.15
I	0 (0)	1 (1)	0 (0)			0.0	0.8	0.0		
II	11 (9)	14 (13)	21 (20)			12.8	12.4	14.6		
III	94 (76)	80 (76)	71 (68)			74.7	76.3	74.3		
IV	19 (15)	10 (10)	13 (12)			12.4	10.5	11.1		
Tumor length, cm	6.0 ± 2.8	5.5 ± 2.6	5.4 ± 3.0	0.17	0.22	5.7 ± 2.7	5.5 ± 2.6	5.6 ± 2.8	0.59	0.08
Tumor location				0.85	0.02				0.77	0.11
Proximal	26 (21)	22 (21)	23 (22)			19.3	22.1	23.6		
Middle	61 (49)	45 (43)	50 (48)			47.2	43.7	44.7		
Distal	37 (30)	38 (36)	32 (30)			33.5	34.2	31.6		
Years of nCRT				<0.001	0.53				<0.001	0.29
2010–2012	3 (2)	18 (17) ^a^	25 (23) ^a^			4.5	16.3 ^a^	16.3 ^a^		
2013–2015	41 (33)	22 (21)	31 (30)			31.2	22.3	27.1		
2016–2018	80 (65)	65 (62)	49 (47) ^a^			64.3	61.4	56.6		

Abbreviations: GBM-IPTW, generalized boosted modeling-inverse probability of treatment weighting, MASD, maximum absolute standardized difference; BMI, body mass index; ACCI, age-adjusted Charlson’s comorbidity index, nCRT, neoadjuvant chemoradiotherapy. ^a^ Statistically significant difference versus the CROSS group after application of the Bonferroni’s correction for multiple comparisons. * Variable not included in the calculation of propensity score. Data are presented as means ± standard deviations or frequencies (percentages), as appropriate.

**Table 2 cancers-14-02610-t002:** Preoperative outcomes of patients in the GBM-IPTW cohort.

Variable	CROSS	PF5040	PF4500	*p* Value
Chemotherapy completion rate (>80%)	89.4	76.2 ^a^	92.0 ^b^	<0.001
Radiotherapy dose, cGy	4401 ± 161	4957 ± 247 ^a^	4426 ± 149 ^b^	<0.001
Weight loss, %	−1.5 ± 5.8	−3.0 ± 9.4	−3.4 ± 8.4 ^a^	0.014
Weight loss >10%	8.5	16.5 ^a^	15.8 ^a^	0.01
Surgical resection rate	76.6	72.8	69.2	0.17
Reason for not undergoing surgery				
Disease progression	8.8	15.6	11.4	0.054
Patient refusal	7.3	9.0	12.7	0.13
Poor physical conditions	5.7	2.0	4.0	0.1
Death during nCRT	1.5	0.6	2.6	0.18

Abbreviations: GBM-IPTW, generalized boosted modeling-inverse probability of treatment weighting; nCRT, neoadjuvant chemoradiotherapy. ^a^ Statistically significant difference versus the CROSS group after application of the Bonferroni’s correction for multiple comparisons. ^b^ Statistically significant difference versus the PF5040 group after application of the Bonferroni’s correction for multiple comparisons. Data are presented as means ± standard deviations or percentages, as appropriate.

**Table 3 cancers-14-02610-t003:** Perioperative outcomes and pathological findings of patients in the GBM-IPTW cohort.

Variable	CROSS	PF5040	PF4500	*p* Value
Time from termination of nCRT to surgery, days	65 ± 31	61 ± 25	65 ± 18	0.16
Thoracic approach				0.2
Thoracotomy	5.0	2.5	2.0	
Thoracoscopy	95.0	97.5	98.0	
Abdominal approach				<0.001
Laparotomy	10.8	31.5 ^a^	35.5 ^a^	
Laparoscopy	89.2	68.5 ^a^	64.5 ^a^	
Type of resection				0.004
Ivor Lewis	9.4	6.5	1.5 ^a^	
McKeown	90.6	93.5	98.5 ^a^	
Postoperative complications				
Anastomotic leak	9.7	24.5 ^a^	27.9 ^a^	<0.001
Chylothorax	1.4	6.7 ^a^	5.6	0.026
Pulmonary	14.1	11.3	26.6 ^ab^	<0.001
Complication severity (Clavien-Dindo)				0.001
None	45.7	44.7	29.3 ^ab^	
Minor (1-3a)	42.3	35.2	52.7 ^b^	
Major or death (3b-5)	12.0	20.1	18.0	
Postoperative stay, days	20.4 ± 15.2	24.2 ± 18.7	25.1 ± 17.8 ^a^	0.019
30-day mortality rate	3.2	2.4	4.9	0.47
ypT stage				<0.001
T0	31.8	48.6 ^a^	38.1	
T1	14.3	12.5	7.6	
T2	13.2	13.0	22.8 ^ab^	
T3	39.6	22.7 ^a^	21.4 ^a^	
T4	1.1	3.3	10.0 ^ab^	
ypN stage				<0.001
N0	76.8	76.5	64.6 ^ab^	
N1	18.3	18.0	23.4	
N2	1.8	5.6	11.2 ^a^	
N3	3.1	0.0 ^a^	0.8	
ypM stage				0.002
M0	100.0	94.9 ^a^	96.5 ^a^	
M1	0.0	5.1 ^a^	3.5 ^a^	
Number of dissected nodes	27.3 ± 12.8	18.4 ± 10.1 ^a^	23.0 ± 10.6 ^ab^	<0.001
Pathologically positive nodes	0.64 ± 2.02	0.45 ± 1.09	0.83 ± 1.59	0.078
ypCR	29.5	45.3 ^a^	31.6 ^b^	0.002
ypT0N+	3.0	3.2	6.5	0.2
Surgical radicality				0.52
R0	91.1	90.8	87.5	
R+	8.9	9.2	12.5	
Tumor regression grade				<0.001
TRG1	31.8	48.6 ^a^	38.1	
TRG2	22.3	17.2	26.8	
TRG3	24.4	28.4	23.7	
TRG4	21.5	5.8 ^a^	11.4 ^a^	

Abbreviations: GBM-IPTW, generalized boosted modeling-inverse probability of treatment weighting; nCRT, neoadjuvant chemoradiotherapy; pCR, pathological complete response; TRG, tumor regression grade. ^a^ Statistically significant difference versus the CROSS group after application of the Bonferroni’s correction for multiple comparisons. ^b^ Statistically significant difference versus the PF5040 group after application of the Bonferroni’s correction for multiple comparisons. Data are presented as means ± standard deviations or percentages, as appropriate.

**Table 4 cancers-14-02610-t004:** Subgroup analysis of overall survival in the PF5040 versus CROSS groups, stratified according to pre-specified variables.

Subgroup	HR (95% CI) PF5040 versus CROSS	*p* for Interaction
ACCI score		0.19
0–2	1.29 (1.001–1.66)	
≥3	0.92 (0.60–1.42)	
Tumor length		0.22
<8 cm	1.10 (0.86–1.40)	
≥8 cm	1.54 (0.95–2.51)	
Years of nCRT		0.5
2010–2014	1.03 (0.71–1.51)	
2015–2018	1.21 (0.93–1.59)	
Chemotherapy completion		<0.001
No	3.81 (2.11–6.89)	
Yes	0.88 (0.69–1.13)	
Surgical resection		0.33
No	1.36 (0.93–1.99)	
Yes	1.08 (0.83–1.41)	
Number of resected nodes		0.29
<15	0.83 (0.45–1.51)	
≥15	1.20 (0.87–1.64)	

Abbreviations: HR, hazard ratio; CI, confidence interval; ACCI, age-adjusted Charlson’s comorbidity index, nCRT, neoadjuvant chemoradiotherapy.

## Data Availability

Data supporting reported results can be provided upon request.

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
