# Peer review of "Neoadjuvant Carboplatin/Paclitaxel versus 5-Fluorouracil/Cisplatin in Combination with Radiotherapy for Locally Advanced Esophageal Squamous Cell Carcinoma: A Multicenter Comparative Study"

_cancers, 2022, doi:10.3390/cancers14112610_

Round 1

Reviewer 1 Report

This study focused on a retrospective comparison of efficacy and safety between preoperative CBDCA+PTX+RT and CDDP+5-FU+RT (50.4Gy), CDDP+5-FU+RT (45Gy) for resectable locally advanced ESCC patients. However, this study included resectable locally advanced ESCC and borderline unresectable locally advanced ESCC patients as the authors mentioned. Therefore, this study might not be able to conclude and discuss optimal preoperative treatment for resectable locally advanced ESCC due to mixed analyzing different populations which had different prognoses.

  • The definition of RFS is inadequate. Author defined the RFS as the time from the R0 or R1 resection to the recurrence and/or death. With this definition, the patients who did not undergo surgery and did not achieve R0/1 resection were omitted by the RFS analysis. The starting time should be defined as the time of initiation of the nCRT and event should be defined as the progression (PFS).
  • Although IPTW is said to have aligned the patient backgrounds, there is a large difference in the percentage of completed preoperative treatment between PF5040 and PF4500 (76.2% vs. 92.0%). Is there such a large difference in the percentage of preoperative treatment completion due to the difference in doses? There may be factors in the patient background that have not been adjusted for.
  • I have the impression that the difference in postoperative complications is directly related to the difference in OS.

The number of lymph nodes resected was different in each group, and we should consider that more lymph nodes were dissected in the CROSS regimen, which may have contributed to survival. It is possible that the CROSS regimen and FP-RT 45 Gy also have slightly different radiation doses, and that this may have resulted in fewer nodes being resected.

Reviewer 2 Report

The data would be meaningful, while reconsidered points are also raised.

  1. The relevance of CROSS on clinical outcomes of esophageal cancer has been previously suggested. It would be required to emphasize more the novel relevance of the study findings in such conditions.
  2. More explanations of Asia-specific features of esophageal cancer, in relation to its epidemiology, pathology and clinics, are required in comparison to Western countries.
  3. The rightfulness/validity of sample size can be demonstrated.
  4. The study deemed to be for Taiwanese patients, although the description of text included both expressions of Asian and Taiwanese patients. More precise description on studied populations can be required.
  5. The rightfulness/validity of covariables that were used for propensity score matching can be more discussed. Comorbidities such as diabetes, lung disorders and heart diseases or additional conditions such as operative hours in a surgical period as a prognosis-related factor may be considered in clinical settings.
  6. Abstract (page 1): the underline between Abstract: and Background: must be deleted.
  7. Abstract: CI ‘-‘ might be described as CI ‘=’.
  8. Abstract: the phrase ‘in Asia or Taiwanese patients’ can be needed in the last sentence.
  9. Methods: the definition of ‘high-volume centers can be required because the definitions differ across Asian countries.
  10. Statistical analysis: the sentence of a value <0.2 as a negligible difference can be required to have an adequate reference.
  11. Results and discussion: the results of subgroup analysis after propensity score analysis may be carefully interpretated.
  12. Page 10: many readers may not understand the word ‘Cf’.
  13. Page 10: the underline in Acknowledgements must be deleted.
  14. Page 10: Authors’ contributions must be expressed in bold.
  15. Overall text: the rule of decimal points must be unified; for instance, 45 Gy or 45.0 Gy. 92% or 92.0%. P-values in some parts.
  16. Native check is further required.

Author Response

AUTHORS’ RESPONSES TO COMMENTS FROM REVIEWER #2

  1. The relevance of CROSS on clinical outcomes of esophageal cancer has been previously suggested. It would be required to emphasize more the novel relevance of the study findings in such conditions.

Response: Thank you for your valuable contribution.  We have commented on this issue in the revised “Introduction” section, as follows: “Despite the promising results obtained for the subset of patients with esophageal squamous cell carcinoma (ESCC) in the CROSS trial (i.e., pathologically complete re-sponse [pCR] rate = 49%; median survival = 81.6 months), replication of these findings in Asian patients with ESCC has not been successful.13−15 While the limited toxicity profile of CROSS is commonly recognized, nCRT in Asia is frequently reserved to patients with  more advanced malignancies (cT3-4aN1-3M0).16-18 However, the efficacy of CROSS in advanced disease stages has been recently called into question by a study from Hong Kong – which showed an unfavorable survival trend for patients who received CROSS versus those who had been treated with PF.13 In addition, traditional PF schemes with 41.4-50.4 Gy of RT are still extensively used for nCRT in Asian patients with EC3”.

  1. More explanations of Asia-specific features of esophageal cancer, in relation to its epidemiology, pathology and clinics, are required in comparison to Western countries.

Response: Thank you for your pertinent comment. We have addressed this concern by revising the beginning of the “Discussion” section, as follows: “Prior investigations focusing on the safety and efficacy of the CROSS regimen have been mainly conducted in Western countries and the results might not be generalizable to different ethnicities. Additionally, Asian patients are generally offered nCRT only in presence of advanced tumors (cT3-4aN1-3M0). This practice is different from that im-plemented in Western countries, where all locally advanced resectable tumors (cT1N1-3M0 / cT2-4aN0-3M0) are treated with nCRT prior to surgery.16-18 To our knowledge, no prospective study in Asia has addressed the effectiveness of CROSS and retrospective studies in the field have been inconclusive”.

  1. The rightfulness/validity of sample size can be demonstrated.

Response: Thank you for pointing this out. First, all eligible patients according to the inclusion and exclusion criteria were enrolled in the study. This has been clarified in the “Methods” section. In addition, we commented on the lack of a formal sample size calculation in the revised “Discussion” section, as follows: “Finally, we did not conduct a formal sample size calculation; therefore, randomized controlled trials are necessary to confirm our findings and to evaluate the impact of different nCRT regimens in terms of survival and complications for Asian patients with ESCC”.

  1. The study deemed to be for Taiwanese patients, although the description of text included both expressions of Asian and Taiwanese patients. More precise description on studied populations can be required.

Response: We completely agree with your comment. In the “Methods” and “Results” section have changed “Asian” to “Taiwanese”. When commenting on the CROSS study, we maintained the reference to Asians since the general lack of evidence was not limited to Taiwan but it also appliable to China, Japan, and other Asian countries.

  1. The rightfulness/validity of covariables that were used for propensity score matching can be more discussed. Comorbidities such as diabetes, lung disorders and heart diseases or additional conditions such as operative hours in a surgical period as a prognosis-related factor may be considered in clinical settings.

Response:  Among the baseline variables listed in Table 1, those identified as independent predictors of survival were subjected to PS matching. This methodology is in line with a previous study, which has been added to the reference list (reference 30). This point has been clarified in the revised “Statistical analysis” section.

  1. Abstract (page 1): the underline between Abstract: and Background: must be deleted.

Response: Thank you for highlighting this. This has now been amended.

  1. Abstract: CI ‘-‘ might be described as CI ‘=’.

Response: Thank you for pointing this out. The sentence has been modified accordingly.

  1. Abstract: the phrase ‘in Asia or Taiwanese patients’ can be needed in the last sentence.

Response: Thank you. This has now been modified by mentioning Taiwanese patients.

  1. Methods: the definition of ‘high-volume centers can be required because the definitions differ across Asian countries.

Response: In Taiwan, a high-volume center is defined as an institution that performs >20 esophagectomies per year. This has been clarified in the revised manuscript.

  1. Statistical analysis: the sentence of a value <0.2 as a negligible difference can be required to have an adequate reference.

Response: Thank you for your comment. As per your request, we have added the following reference: McCaffrey, D.F.; Griffin, B.A.; Almirall, D.; Slaughter, M.E.; Ramchand, R.; Burgette, L.F. A tutorial on propensity score estimation for multiple treatments using generalized boosted models. Statistics in Medicine 2013, 32, 3388-3414, doi:10.1002/sim.5753.

  1. Results and discussion: the results of subgroup analysis after propensity score analysis may be carefully interpretated.

Response: In the IPTW method, weights are assigned according to the baseline variables in the original cohort. When subgroup analyses were implemented, weights were not reassigned and the presence of an adequate balance cannot be guaranteed. Therefore, no definitive conclusions can be drawn from these results, which should be merely considered as hypothesis-generating.

  1. Page 10: many readers may not understand the word ‘Cf’.

Response: Thank you for highlighting this. This has now been amended.

  1. Page 10: the underline in Acknowledgements must be deleted.

Response: Thank you for pointing this out. The tex has been modified accordingly.

  1. Page 10: Authors’ contributions must be expressed in bold.

Response: Thank you for highlighting this. This has now been amended.

  1. Overall text: the rule of decimal points must be unified; for instance, 45 Gy or 45.0 Gy. 92% or 92.0%. P-values in some parts.

Response: We apologize for this error when transcribing the data. As per your request, we have revised all numbers and decimal points.

  1. Native check is further required.

Response: Thank you. The manuscript has been revised for style and presentation.

Reviewer 3 Report

Excellent paper using a sophisticated methodology to overcome the shortcomings of a retrospective study. This paper strongly support the use of neoadjuvant chemoradiotherapy CROSS for advanced ESSC also in Eastern populations.

Author Response

Response: We greatly appreciate the reviewer’s supportive comment on our manuscript.

Round 2

Reviewer 2 Report

The parts suggested by this reviewer have been improved. As one more request (if possible), in Introduction, the authors may describe more about whether or not the clinic-epidemiological features of esophageal cancers (i.e. occurrence sites [upper, middle, lower], histopathology, gender, genetics, etc.) can differ among Taiwan, other Asian, or Western people because lifestyles (e.g. alcohol, smoking, tea), obese traits, and the prevalence of GERD (efflux syndrome) differ among these countries. Thank you.

Author Response

Thank you for your comments! With your suggestions for our "Introduction" session, we have amended this part: "In Asia, more than 90% of all EC diagnosed are esophageal squamous cell carcinoma (ESCC). Not only do risk factors differ from western regions (i.e. hot drinks, foods containing N-nitroso compounds), varying responses to anticancer therapies between Asian and Caucasian patients has also been linked to varying frequencies of genetic polymorphisms.16,17 Furthermore, neoadjuvant treatment is reserved to patients with more advanced malignancies (cT3-4aN1-3M0) in Asia.18  The efficacy of CROSS in advanced stages of ESCC has recently been called into question by a study from Hong Kong – which showed an unfavorable survival trend for patients who received CROSS versus those who had been treated with PF.13 In addition, traditional PF schemes with 41.4-50.4 Gy of RT are still extensively used for nCRT in Asian patients with EC.3"